# Citrus Auraptene Induces Expression of Brain-Derived Neurotrophic Factor in Neuro2a Cells

**DOI:** 10.3390/molecules25051117

**Published:** 2020-03-03

**Authors:** Yoshiko Furukawa, Yu-suke Washimi, Ryu-ichi Hara, Mizuki Yamaoka, Satoshi Okuyama, Atsushi Sawamoto, Mitsunari Nakajima

**Affiliations:** Department of Pharmaceutical Pharmacology, College of Pharmaceutical Sciences, Matsuyama University, 4-2 Bunkyo-cho, Matsuyama, Ehime 790-8578, Japan; 16151100@g.matsuyama-u.ac.jp (Y.-s.W.); 16140866@g.matsuyama-u.ac.jp (R.-i.H.); mu.yakuri.016@gmail.com (M.Y.); sokuyama@g.matsuyama-u.ac.jp (S.O.); asawamot@g.matsuyama-u.ac.jp (A.S.); mnakajim@g.matsuyama-u.ac.jp (M.N.)

**Keywords:** auraptene, brain-derived neurotrophic factor, BDNF, neuro2a cells, neuroprotective effect

## Abstract

(1) Background: Our published data have indicated that (1) auraptene (AUR), a citrus ingredient, has neuroprotective effects on the mouse brain, owing to its ability to suppress inflammation, such as causing a reduction in hyperactivation of microglia and astrocytes; (2) AUR has the ability to trigger phosphorylation (activation) of extracellular signal-related kinase (ERK) and cAMP response element-binding protein (CREB) in neuronal cells; (3) AUR has the ability to induce glial cell line-derived neurotrophic factor (GDNF) synthesis/secretion in rat C6 glioma cells. The well-established fact that the ERK-CREB pathway plays an important role in the production of neurotrophic factors, including GDNF and brain-derived neurotrophic factor (BDNF), prompted us to investigate whether AUR would also have the ability to induce BDNF expression in neuronal cells. (2) Methods: Mouse neuroblastoma neuro2a cells were cultured and the effects of AUR on BDNF mRNA expression and protein content were evaluated by RT-PCR and ELISA, respectively. (3) Results: The levels of BDNF mRNA and secreted BDNF were significantly increased by AUR in a dose- and time-dependent manner in neuro2a cells. (4) Conclusion: The induction of BDNF in neuronal cells might be, in part, one of the mechanisms accounting for the neuroprotective effects of AUR.

## 1. Introduction

Auraptene (7-geranyloxycoumarin; AUR), a citrus coumarin derivative, has been revealed to possess valuable pharmacological properties, including anti-carcinogenic, anti-inflammatory, anti-oxidative, anti-helicobacter, anti-diabetic, anti-hypertensive [1], and neuroprotective ones [2]. We previously demonstrated that AUR exerts anti-inflammatory effects in not only peripheral organs but also the brain, as found from studies using a mouse model of global cerebral ischemia [3,4] and lipopolysaccharide (LPS)-induced systemic inflammation [5,6]. In the process of studying about the neuroprotective effects of AUR, we clarified that (1) AUR has the ability to phosphorylate (i.e., activate) extracellular signal-related kinase (ERK), a component of mitogen-activated protein kinase (MAPK), in rat primary cortical neurons [7]; (2) In the rat pheochromocytoma cell line (PC12 cells), AUR can cause phosphorylation of ERK and of cAMP response element-binding protein (CREB), a downstream target of ERK and a crucial transcription factor for neuronal plasticity and long-term potentiation (LTP) in the brain [8]; and (3) AUR phosphorylates ERK and CREB in rat C6 glioma cells [9]. It is well known that the ERK-CREB pathway plays an important role in the production of neurotrophic factors, such as glial cell line-derived neurotrophic factor (GDNF) [10] and brain-derived neurotrophic factor (BDNF) [11,12]. In fact, our latest study revealed that AUR has the ability to induce GDNF expression in C6 cells by triggering the protein kinase A (PKA)/ERK/CREB pathway [9]. Using neuro2a cells, an immortalized cell line of murine nerve cells, we thus examined herein whether AUR would have the ability to induce BDNF expression in these cells.

## 2. Results

### 2.1. Effects of AUR on the Viability of Neuro2a Cells

In order to avoid unfavorable toxic damage to the cells and to confirm the optimal concentration of AUR for the experiments, we treated neuro2a cells for 20 h with various concentrations of AUR (0–80 μM) and then examined its effect on cell viability by performing the MTT assay. Figure 1 shows that there were no significant differences among the colorimetric values for assessing cell metabolic activity (dotted bars) during the treatment with 0–50 μM AUR for 20 h. However, a decrease in values was observed when the cells were treated for 20 h with AUR at the concentration of 60 μM (** *p* < 0.01) or 80 μM (** *p* < 0.01).

When the cells were treated for 40 h with AUR (0–50 μM), the results of the MTT assay showed no significant differences in the colorimetric values (hatched bar) between non-treated cells and those treated with AUR during the treatment. The effect of AUR at the concentration above 60 μM was not considered in 40 h treatment experiment. Based on these results, 10–50 μM AUR was used in subsequent experiments.

### 2.2. Effects of AUR on BDNF mRNA Expression in Neuro2a Cells

Next, we investigated the effect of AUR on *bdnf* gene expression in neuro2a cells. The cells were treated with 10 μM AUR for 0–50 h, and then the extracted total RNA was applied for RT-PCR. As shown in Figure 2A, AUR started to enhance the level of BDNF mRNA after 20 h of treatment (* *p* < 0.05), and the level continued to increase up to 50 h (** *p* < 0.01). These results indicated that the exposure of the cells to 10 μM AUR caused the elevation of BDNF mRNA in a time-dependent manner.

We then treated the cells with 10 or 20 μM AUR for 20 h (Figure 2B) or 5 or 10 μM AUR for 40 h (Figure 2C). In this way, BDNF mRNA almost significantly increased by AUR-treatment in a dose- and time-dependent manner.

### 2.3. Effects of AUR on BDNF Content of Medium Conditioned by Neuro2a Cells

To examine whether AUR treatment induced BDNF protein, we determined BDNF release into the culture medium of neuro2a cells. When the cells were treated with AUR (0–50 μM) for 30 h (Figure 3A), the amount of BDNF released from the neuro2a cells was low at low concentrations of AUR but significantly increased at the concentration of 50 μM (* *p* < 0.05). After treatment for 50 h, 10 μM AUR slightly (*p* = 0.056) increased the BDNF content and 15 μM AUR significantly (*** *p* < 0.001) increased it (Figure 3B). These results showed that AUR induced BDNF release in an almost time- and dose-dependent manner.

We also investigated the intracellular level of BDNF by immunoblot analysis at 30, 50, or 70 h after the AUR treatment. As a result, we observed that both mature BDNF contents and pro BDNF contents in the cell lysates of AUR-treated cells were not different from those of the non-treated cells (data not shown), probably because the sensitivity of immunoblot analysis is rather lower than that of ELISA.

### 2.4. Effects of MEK Inhibitor on AUR-Induced BDNF mRNA Expression in Neuro2a Cells

To examine whether the activation of ERK was responsible for BDNF expression, we pretreated the cells with 10 μM U0126 (a highly selective inhibitor of both MEK1 and MEK2, types of MAPK/ERK kinase) for 30 min and subsequently treated them with 20 μM AUR for 20 h. Figure 4 shows that AUR-induced BDNF mRNA expression was significantly (^#^
*p* < 0.05) inhibited by U0126, indicating that the action of AUR on BDNF synthesis in neuro2a cells was mediated in an ERK-dependent manner.

## 3. Discussion

In this study, we revealed that AUR acted as an inducer of BDNF in neuro2a cells. This finding suggests that the induction of BDNF in neurons might be one of the mechanisms accounting for the neuroprotective effects of AUR in the brain.

A number of recent studies showed that glial cells might be a cellular target for therapeutic approaches to treat several neurodegenerative diseases and neurological disorders, because glial cells, especially astrocytes, serve as key elements in the formation, maintenance, and refinement of synapses [13,14,15]. In our earlier studies regarding the neuroprotective ability of AUR, we revealed that (1) AUR has the ability to suppress inflammatory responses in vivo, namely, hyperactivation of microglia/astrocytes, and over-expression of inflammatory factors by astrocytes, in some mouse models of brain disorders; thus, resulting in the suppression of neuronal death in the hippocampus [3,4,5,6]; and (2) AUR has the ability to induce GDNF expression in C6 cells [9] in vitro. This series of our studies thus suggested that the site of neuroprotective ability of AUR might be astrocytes.

The present findings suggested that some of the neuroprotective effects exerted by AUR could, in part, have resulted from enhanced synthesis/secretion of BDNF by neurons themselves. The amount of BDNF secreted by neuro2a cells was very low (a few pg/mL; Figure 3), whereas that of GDNF secreted by C6 cells was about 10 times as high (tens of pg/mL) [9], probably depending upon the difference in their mechanism of action; BDNF might be neuroprotective in an autocrine manner, and GDNF, in a paracrine one. As previous reports indicated that astrocytes [16] and C6 cells [12] were able to synthesize BDNF, we investigated whether AUR could induce BDNF expression in C6 cells. As a result, we found that AUR induced BDNF mRNA expression but not secretion of the BDNF protein (data not shown). These findings suggested the possibility that neurons and astrocytes secrete each neurotrophic factor in response to AUR.

BDNF has been established as the main neurotrophic factor in the brain [17]. BDNF in the mature brain is known to be stored in neurons and released from neurons in a use-dependent manner [18] for LTP and learning/memory. In addition to these neurotrophic effects, BDNF also possesses neuroprotective effects including anti-apoptosis, anti-oxidation, and autophagy-suppressing ones [19], leading to the expectation that BDNF may have therapeutic potential to treat age-related neurological disorders and neurodegenerative diseases, such as Alzheimer’s disease, Parkinson’s disease, and Huntington’s disease [20]. We should prove in in vivo study the inference that neuron-derived BDNF plays an important role for neuroprotective ability of AUR, using anti-BDNF antibody. However, this proof might be difficult, because AUR is likely to much more powerfully act on glial cells than neurons.

In conclusion, our series of studies suggested that AUR might be a potent neuroprotective citrus ingredient because it has anti-inflammatory action and the ability to induce neurotrophic factors, GDNF in astrocytes, and a little amount of BDNF in neurons. In the future, we would like to investigate the effect of AUR on microglial cells, for example, using N9 cells. The peel of *Citrus grandis* (*Kawachi bankan*), a rich source of AUR [21], might become a natural medicine in the future, as suggested previously [22,23,24,25,26,27].

## 4. Materials and Methods

### 4.1. Chemical and Reagents

AUR was kindly supplied by Ushio ChemiX Corporation (Omaezaki, Japan). AUR was dissolved in dimethyl sulfoxide (DMSO). The final concentration of DMSO in all culture media was below 0.1%.

### 4.2. Cell Culture

Neuro2a cells were maintained and cultured, as previously described [28,29]. Dulbecco’s Modified Eagle Medium (DMEM), fetal bovine serum (FBS), and antibiotics were purchased from Thermo Fisher Scientific (Waltham, MA, USA).

### 4.3. Determination of Cell Viability

Cells seeded in a 96-well plate at a density of 1 × 10^4^ cells/well were cultured for 24 h in a medium containing 10% FBS, then for a further 24 h in medium containing 2% FBS, after which the cells were treated with samples in a medium containing 2% FBS for 20 or 40 h. Cell viability was determined by the MTT assay, as previously described [9,28].

### 4.4. Total RNA Extraction and RT-PCR

Cells seeded in 35-mm culture dishes at a density of 2.5 × 10^5^ cells/dish were cultured for 24 h in a medium containing 10% FBS, then for a further 24 h in a medium containing 2% FBS, after which the cells were treated with samples in a medium containing 2% FBS for the indicated times. Total RNA from neuro2a cells was prepared by use of Isogen II (Nippon Gene, Tokyo, Japan), as previously described [9]. Reverse transcription (RT) and PCR were carried out with a Takara PrimeScript^TM^ RT-PCR kit (Takara Bio Inc., Kyoto, Japan). The following primer pairs were used: 5′-CCC AGG GCA GGT TCG AGA GG -3′ (20 mer) and 5′-CCC GCC AGA CAT GTC CAC TG -3′ (20 mer) for BDNF; 5′-CGG AGT CAA CGG ATT TGG TCG TAT-3′ (24 mer) and 5′-AGC CTT CTC CAT GGT GGT GGA GAC-3′ (24 mer) for glyceraldehyde-3-phosphate dehydrogenase (GAPDH). The number of PCR cycles and specific annealing temperatures were 34 cycles and 61°C for BDNF, and 23 cycles and 55 °C for GAPDH. RT-PCR is the “end-point PCR” and semiquantitative method. The number of samples was 3 of different culture, and the number of measurements was 3 for each sample.

### 4.5. ELISA

Cells seeded in 35-mm culture dishes at a density of 2.5 × 10^5^ cells/dish were cultured for 24 h in a medium containing 10 % FBS, then for a further 24 h in a medium containing 2 % FBS, after which the cells were treated for the indicated times with samples in medium containing 2 % FBS. The culture media were centrifuged at 2000× g for 20 min to remove particulates before analysis. BDNF contents in cell culture supernatants were measured by using a biosensis^®^ Mature BDNF Rapid ELISA Kit (Biosensis Pty Ltd., Thebarton, South Australia). The typical lower limit for BDNF detection is 1–2 pg/mL.

### 4.6. Statistical Analysis

All results were expressed as means ± SEM. Significant differences of experiments with 2 groups were analyzed by performing Student’s *t*-test. Experiments with 3 or more groups were subjected to a one-way ANOVA, followed by Dunnet’s multiple comparison test. *p* < 0.05 was taken to be statistically significant.

## Figures and Tables

**Figure 1 molecules-25-01117-f001:**
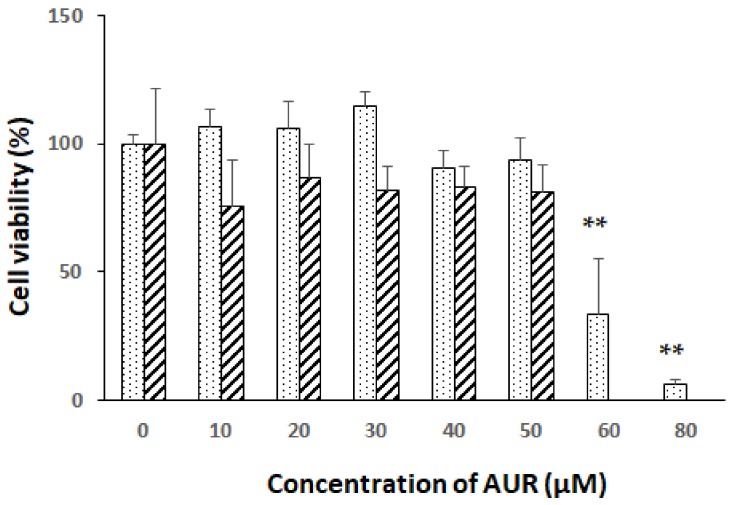
Effects of treatment with auraptene (AUR) on neuro2a cell viability. Cells were treated with various concentrations (10–80 μM) of AUR for 20 h (dotted bars) or with various concentrations (10–50 μM) of AUR for 40 h (hatched bar). The results represent the mean ± SEM (n = 4, different culture). Significance difference in values between the non-treated and AUR-treated cells: ** *p* < 0.05.

**Figure 2 molecules-25-01117-f002:**
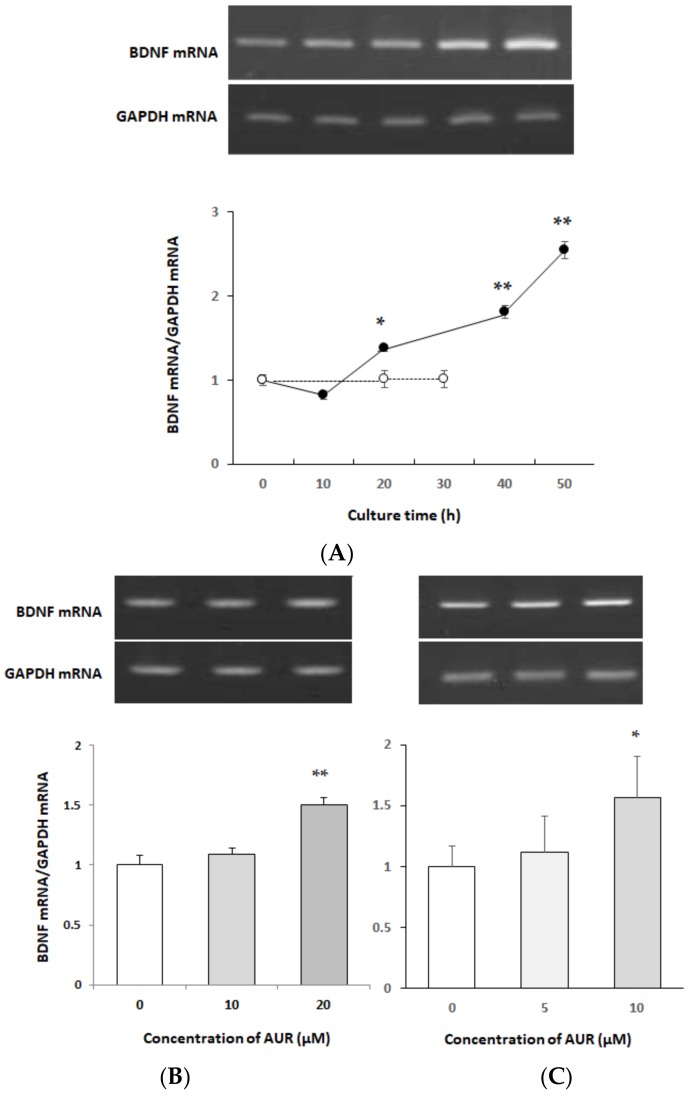
Effects of treatment with AUR on brain-derived neurotrophic factor (BDNF) mRNA content in neuro2a cells. (**A**) Cells were incubated with (●) or without (○) 10 μM AUR for various times (0–50 h). (**B**) Cells were incubated with 10 or 20 μM AUR for 20 h. (**C**) Cells were incubated with 5 or 10 μM AUR for 40 h. Total RNA levels of untreated cells and of those treated with AUR were analyzed by the RT-PCR method. The results represent the mean ± SEM (n = 3, different culture). Significance difference in values between the non-treated and AUR-treated cells: * *p* < 0.05; ** *p* < 0.01.

**Figure 3 molecules-25-01117-f003:**
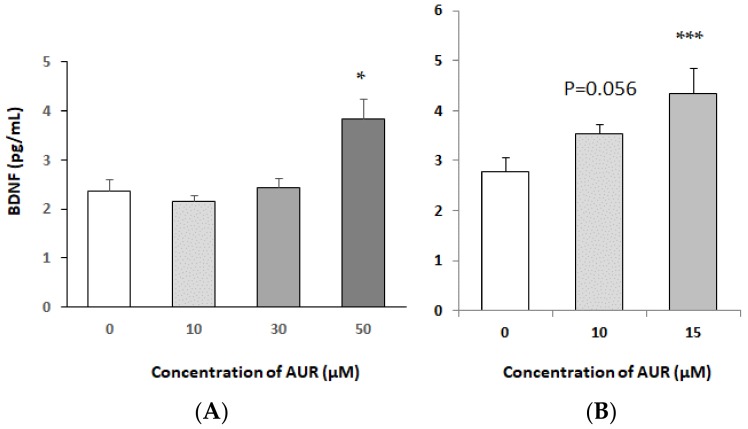
Effects of treatment with AUR on BDNF content in medium conditioned by neuro2a cells. (**A**) Cells were incubated with 10, 30, or 50 μM AUR for 30 h. (**B**) Cells were incubated with 10 or 15 μM AUR for 50 h. The results represent the mean ± SEM (n = 3, different culture). Significance difference in values between the non-treated and AUR-treated cells: * *p* < 0.05; *** *p* < 0.001.

**Figure 4 molecules-25-01117-f004:**
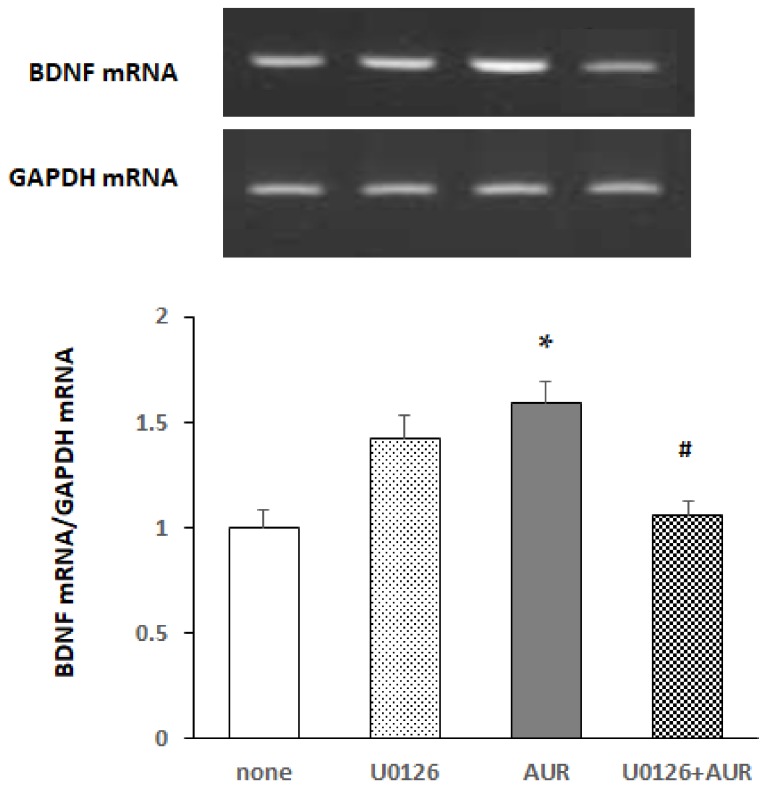
Effect of U0126 on AUR-induced increase in BDNF mRNA content in neuro2a cells. Cells were preincubated with or without 10 μM U0126 for 30 min and then incubated with 20 μM AUR for 40 h. The results represent the mean ± SEM (n = 3, different culture). Significance difference in values between the non-treated and AUR-treated cells: * *p* < 0.05; significant difference in values between the AUR-treated and AUR/inhibitor-treated cells: ^#^
*p* < 0.05.

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
