# Peer review of "Citrus Auraptene Induces Expression of Brain-Derived Neurotrophic Factor in Neuro2a Cells"

_molecules, 2020, doi:10.3390/molecules25051117_

Round 1
Reviewer 1 Report
The authors submitted the MS “Citrus auraptene induces expression of brain-derived neurotrophic factor in neuro2a cells.” By Yoshiko Furukawa*, Yu-suke Washimi, Ryu-ichi Hara, Mizuki Yamaoka, Satoshi Okuyama, Atsushi Sawamoto, and Mitsunari Nakajima. In introduction indicated that Auraptene (7-geranyloxycoumarin; AUR), a citrus coumarin derivative, possess valuable pharmacological properties, including anti-carcinogenic, anti-inflammatory, anti-oxidative, anti-helicobacter, anti-diabetic, anti-hypertensive and neuroprotective ones. According to their OUR effects, the authors postulated an aim, whether AUR would have the ability to induce brain-derived neurotrophic factor (BDNF) expression in neuro2a murine nerve cells. The results show that AUR could induce BDNF expression in neuronal cells. Since AUR acted as an inducer of BDNF in neuro2a cells, suggesting that the induction of BDNF in neurons might be one of the mechanisms for the neuroprotective effects of AUR in the brain. The main results show that “BDNF mRNA almost significantly increased by AUR-treatment in a dose- and time-dependent manner. The AUR treatment also induced BDNF protein release into the culture medium of neuro2a cells in an almost time- and dose-dependent manner. By studying the effects of MEK inhibitor on AUR-induced BDNF mRNA expression in neuro2a cells, the results shows that AUR-induced BDNF mRNA expression was significantly (#p < 0.05) inhibited by U0126, indicating that the action of AUR on BDNF synthesis in neuro2a cells was mediated in an ERK-dependent manner. After the review, I suggest accepting for publication at the Journal, the results are interesting for the audience.
Author Response
The answer to Reviewer 1:
I would like to express my deep appreciation to kind review of reviewer 1.
Reviewer 2 Report
The manuscript by Nakajima and colleague's describes the use of a citrus auraptene on the expression of BDNF in neuro2a cells. This was measured by RT-PCR and ELISA. Currently, the manuscript shows very preliminary data which lacks novelty, and therefore I cannot recommend publication of this manuscript.
Specific comments.
Overall RT-PCR and not qPCR is not a sufficient method to detect mRNA levels. There was little change in protein secretion as measured by ELISA and total protein of BDNF was not changed in their study (which they did not show), which makes the changes in gene expression insignificant. There are no functional experiments on the effect of AUR in this study. They should show that the neuroprotective effect of AUR is at least partially dependent on BDNF production.Author Response
The answer to Reviewer 2:
According to the kind suggestion of reviewer 2, we collected our manuscript as follows.
1) About “Overall RT-PCR and not qPCR is not a sufficient method to detect mRNA levels.”;
As pointed out by reviewer, RT-PCR is semi-quantitative method. We thus prepared the total RNA samples with 3 different cultures and each sample was measured three times. We explained about this fact in the Materials and Methods (line 260-262).
2) About “There was little change in protein secretion as measured by ELISA and total protein of BDNF was not changed in their study (which they did not show), which makes the changes in gene expression insignificant.”:
The amount of BDNF secreted by neuro2a cells were very low level. But we've confirmed this fact with careful measurement. As they are reproducible data, we have no doubt that “There was a little change in protein secretion”.
We did not find the reports showing the BDNF level secreted by neuro2a cells. The BDNF level secreted by astrocytes has been reported to be a few pg/mL (Ref.16; Toyomoto et al., 2005). Although another group indicated that the BDNF level secreted by astrocytes and cortical neurons are dozens of pg/mL (Jeon et al., Neurosci.Res. (2011) 69: 214-222), we think it is reasonable that neuro2a cells have the ability to secrete only a small amount of BDNF.
As pointed out by reviewer, the expression “In conclusion, our series of studies strongly suggested that AUR might be a potent neuroprotective citrus ingredient because it has anti-inflammatory action and the ability to induce neurotrophic factors, GDNF in astrocytes and BDNF in neurons.” was scarcely too much to say. We thus changed this expression to “In conclusion, our series of studies suggested that AUR might be a potent neuroprotective citrus ingredient because it has anti-inflammatory action and the ability to induce neurotrophic factors, GDNF in astrocytes and a little amount of BDNF in neurons.” (line 229-231).
We also added the sentence “We should prove in in vivo study the inference that neuron-derived BDNF plays an important role for neuroprotective ability of AUR, using anti-BDNF antibody. However, this proof might be difficult, because AUR is likely to much more powerfully act on glial cells than neurons.” (line 225-228).
Reviewer 3 Report
The topic of the paper is interesting and in the last years recent studies showed that glial cells might be a cellular target for therapeutic approaches to treat several neurodegenerative disease.
The induction of BDNF in neuronal cells might be one of the mechanism for the neuroprotective effects of auraptene but would be interesting do another experiments in other cell line (for example N9) for study in good manner the difference in mechanism of action.
For the effect of treatment is better have more than only 3 sample or 4.
129- The results represent the mean and SEM (n=3)
Author Response
The answer to Reviewer 3:
According to the kind suggestion of reviewer 3, we collected our manuscript as follows.
1) About “…would be interesting do another experiments in other cell line (for example N9) for study in good manner the difference in mechanism of action.”;
As pointed out by reviewer, it is interesting to study the effect of auraptene on other types of cells. We have previously investigated the effect of AUR on astrocytes using C6 cells (Ref.9; Furukawa et al., 2019), and mentioned in Introduction (line 46-47) and in Discussion (line 229-231). As for microglia, we would like to investigate the effect of AUR on these cells using N9 cells someday. We added the sentence that “In the future, we would like to investigate the effect of AUR on microglial cells, for example, using N9 cells.” (line 231-232).
2) About “For the effect of treatment is better have more than only 3 sample or 4.”:
As pointed out by reviewer, we have prepared the samples from 3 different culture (n=3) for ELISA and RT-PCR. As RT-PCR is semi-quantitative method, we prepared the total RNA samples with 3 different cultures and each sample was measured three times. We explained about this fact in the Materials and Methods (line 260-262). As for ELISA, we analyzed under various conditions, and we have no doubt that “There was a little change in protein secretion.”.
We changed the description from “n=3” to “n=3, different culture” in the figure legends of Figure 3 (line 163) and Figure 4 (line 216).
Round 2
Reviewer 2 Report
The authors have not sufficiently addressed my previous concerns, and therefore, again, I cannot recommend publication.